# Incidence and Risk Factors for Urinary Tract Infection in an Elder Home Care Population in Taiwan: A Retrospective Cohort Study

**DOI:** 10.3390/ijerph16040566

**Published:** 2019-02-16

**Authors:** Wei-Yi Shih, Chia-Chen Chang, Meng-Ting Tsou, Hsin-Lung Chan, Ying-Ju Chen, Lee-Ching Hwang

**Affiliations:** 1Department of Family Medicine, MacKay Memorial Hospital No. 92, Sec. 2, Zhongshan N. Rd., Zhongshan Dist., Taipei City 104, Taiwan; ginger3726@gmail.com (W.-Y.S.); betty905@gmail.com (C.-C.C.); mttsou@gmail.com (M.-T.T.); hunter5770@gmail.com (H.-L.C.); 2Telehealth and Home Care Center, MacKay Memorial Hospital No. 92, Sec. 2, Zhongshan N. Rd., Zhongshan Dist., Taipei City 104, Taiwan; ying-ju@mmh.org.tw; 3Department of Medicine, MacKay Medical College No. 46, Sec. 3, Zhongzheng Rd., Sanzhi Dist., New Taipei City 252, Taiwan

**Keywords:** home care, urinary tract infection, risk factors

## Abstract

Urinary tract infection (UTI) is a common infection worldwide. Numerous studies have demonstrated risk factors for UTI in facilities and communities. In this study, we aimed to identify risk factors for UTI in home care patients. Patients who received home care for at least three months in 2017 were included. A UTI episode was defined by admission for UTI treatment, and/or a visit to an outpatient department for UTI and receiving antibiotic prescription. A total of 598 patients aged 81.9 years were included. Overall 47% (281) had at least one UTI episode. After analysis, urinary catheter indwelling was the most important risk factor (odds ratio (OR), 3.21). Underlying diseases (coronary artery disease (CAD), chronic kidney disease (CKD), diabetes mellitus (DM)) were related to UTI with OR ranging from 1.46 to 1.79. Higher Alb (albumin) (OR 0.68) and Hb (hemoglobin) (OR 0.91) were inversely related to UTI. Polypharmacy (OR 1.84) and lower Barthel index scores (OR 1.62) were also related to UTI by different degrees. In conclusion, apart from the unmodifiable factors, avoiding prolonged urinary catheter placement, unnecessary prescriptions, and keeping good nutritional status may help to prevent UTI in home care patients.

## 1. Introduction

Life expectancy is substantially extended due to advances in medical sciences and a decline in the fertility rate. According to the United Nations, the number of people aged 60 years and older is expected to increase to 2.1 billion by 2050 [1]. Taiwan has officially entered the stage of an aged society, with more than 14% of the population older than 65 years in 2018, and is estimated to reach the threshold of a super-aged society by 2025 [2]. According to the Report of the Senior Citizen Condition Survey launched by Ministry of Health and Welfare (MOHW) of Taiwan in 2017, among those who were aged 65 years or older, 28.1% (about 907,000) required assistance in at least one domain of basic or instrumental activities of daily living. However, there was only one third of elderly willing to be institutionalized while they became disabled [3].

Home care is a rising care model in Taiwan, in which medical services are provided by medical staff in a community setting. According to the statistics of the National Health Insurance Administration, MOHW, the number of patients receiving home care service was 104,197 in 2017, and this population is expected to grow rapidly after the implementation of the Long-Term Care Service Act 2.0 in 2017 [4]. Thus, this care system and special subgroup have drawn experts’ attention recently [5,6].

The aging process leads to a constellation of physiologic change and accumulation of comorbidities. Compared to younger people, the elderly are more likely to be victims of falls, delirium, depression, and infections. Among various infections, urinary tract infection (UTI) is one of the most common. Its incidence rises substantially in the elderly, which exhibits not only a clinical but also economic burden [7]. Although the estimated mortality rate is generally lower than with respiratory tract infection, it may rise up to 26% if complicated with bacteremia or septic shock [8]. While some studies have reported the risk factors for UTI in the elderly, they mainly focused on residents in facilities [9].

A large-scale study using the 2010 national Outcome and Assessment Information Set data demonstrated that, apart from respiratory tract infection and wound infections, UTI is the 3rd leading cause of infection in home care population, leading to 4.4% of overall unplanned hospitalizations in the United States [6]. An investigation targeting Finnish home care clients in 2014 reported a 30-day UTI prevalence rate of 4.5%, and female gender and urinary catheter were the strongest factors associated with UTIs [10]. Another study in Taiwan also reported factors related to UTI in home care population, such as dementia, urinary catheter, and chronic kidney disease [11]. Nevertheless, the observation periods of previous studies were relatively short and only patients admitted for UTI were enrolled, despite the fact that a large proportion of UTI episodes were treated at outpatient departments. Therefore in this study, we aim to investigate the incidence and factors related to UTIs both in outpatient and inpatient settings, and to propose strategies to prevent UTI and improve the quality of home care.

## 2. Materials and Methods

### 2.1. Home Care System in Taiwan

In Taiwan, home care services are reimbursed by the NHI program. A patient eligible for home care services must have definite medical or nursing care needs, with limited ability of self-care, and have chronic conditions requiring long-term care or continual care needs following hospital discharge. The limited ability of self-care is defined by the patient being chair- or bed-bound over 50% of the time while awake. The services items include general and skilled nursing services, and physician visits.

### 2.2. Participants

The study population came from a home care database of a single medical center in northern Taiwan. To assess the risk factors which would be encountered while receiving home care service rather than acute disease stage, only patients who had been followed up for at least 3 months from 1 January 2017 to 31 December 2017, were enrolled. Those who entered the service system before 2017 and remained under care in 2017 were also included. Residents in facilities were excluded. Information, such as age, sex, source of care giver, functional status, main co-morbidities, presence of urinary catheter, and average serum hemoglobin and albumin levels, were obtained from the database.

### 2.3. Measures

Because most of the patients in this study were completely bed bound, body weight (BW) and length (BL) were estimated using the anthropometric method [12,13]. BW was calculated using the following equations: Male BW = −72.4104 + 1.1228 × hip circumference (HC) + 1.1268 × mid-arm circumference (MAC); Female BW = −51.3536 + 0.8203 × HC + 1.0831 × MAC. 

Body mass index (BMI) was calculated with the following equation: BW/BL2 (kg/m^2^). For body height, the following equations were used: Male body height = 85.10 + 1.73 × knee height (KH) − 0.11 × age; Female body height = 91.45 + 1.53 × KH − 0.16 × age.

The comorbidities in this study were defined using the ICD-10 codes recorded in the medical documents. A UTI episode during the observation period was defined by any one of the following conditions: (1) admission for UTI treatment, and (2) visit to outpatient department for UTI and receiving antibiotic prescription. UTI diagnosis was confirmed by any ICD-10-CM codes recorded in outpatient or admission medical documents as following: Urinary tract infection (N39.0), acute cystitis/acute cystitis without hematuria (N30.0), acute cystitis with hematuria (N30.01), other cystitis without hematuria (N30.8), other cystitis with hematuria (N30.81), cystitis, unspecified without hematuria (N30.9), cystitis, unspecified with hematuria (N30.91), and acute pyelonephritis (N10).

As the study population generally had multiple prescriptions for underlying diseases, we chose taking 10 or more prescriptions over 90 days as the cut-off point, which was termed as hyper-polypharmacy or excessive polypharmacy [14]. Chronic urinary catheter indwelling was defined as documented presence of urinary catheter over 30 days during the observation period and required changing catheter every 2 weeks to 1 month depending on the type of catheter.

The Barthel index score represented the ability of a patient to perform activities of daily living and included the following domains: Toileting, bathing, eating, dressing, continence, transfers, and ambulation. It was assessed at the time of entry to home care service and was classified into totally dependent (scores 0–20) or severely dependent (scores 21–60). The conscious state of an individual was assessed using the Glasgow coma scale (GCS), which was defined as severe impairment (GCS < 8), moderate impairment (GCS 9–12), and minor impairment (GCS ≥ 13).

The average mini-nutritional assessment (MNA) score was used for nutritional status assessment. MNA distinguished patients with adequate nutritional status (MNA ≥ 24), at risk of malnutrition (24 > MNA ≥ 17), and malnourishment (MNA < 17) [15]. Laboratory data, such as average serum hemoglobin and albumin levels, were expressed as mean ± standard deviation.

### 2.4. Data Analysis

Categorical variables were presented as percentage, and statistical analysis was performed using chi-square test. Continuous variables were presented as mean ± standard deviation, and statistical analysis was carried out using two-tailed t test or ANOVA test. The risk factors related to UTI were further assessed by multiple logistic regression analysis by adjusting age and sex. Significance was defined as *p* < 0.05. The SPSS for Windows analytical software (version 24.0) (IBM, Armonk, NY, USA) was used for all analyses.

## 3. Results

After excluding the residents in facilities, 598 patients who received over 3 months of home care service during the observation period were included in this study. The basic demographic characteristics are shown in Table 1. The number of females was 362 (60.5%), and of males, 236 (39.5%). The average age was 81 ± 11.3 years. Family member (53.2%) was the main source of care giver, followed by foreign (44.3%) and Taiwanese workers (2.5%). The mean BMI was 22.0 ± 3.6 kg/m^2^. A high prevalence rate of multiple comorbidities, such as hypertension (79.9%), cerebrovascular disease (50.3%), and diabetes mellitus (DM) (46.5%), were found. About 20% of the patients met the definition of hyper-polypharmacy. Half of them had indwelling urinary catheter (50.3%), and most patients (81.8%) were classified as totally ADL (activities of daily living) dependent. The patients were mostly malnourished, with only four patients classified as well-nourished.

There were 281 patients who had at least one UTI episode either diagnosed at outpatient or inpatient department during the observation period, accounting for an overall incidence of 47%. Of the 281 patients, 220 (78.3%) had been admitted for UTI treatment and 50 died of UTI, resulting in a mortality rate of 22.7%. 

After grouping the population into UTI and non-UTI groups, we found that there was no difference in age, sex, BMI, or source of care giver (Table 2). Underlying diseases, such as coronary artery disease (CAD), chronic kidney disease (CKD), and DM, were more prevalent in the UTI group. A significant difference was also found in the rate of hyper-polypharmacy. The UTI group was more ADL dependent and showed higher urinary incontinence rate and more urinary catheter indwelling rate than the non-UTI group. Moreover, the UTI group had lower hemoglobin and albumin levels, but the difference in MNA scores was not significant.

After adjusting age and sex, the multiple logistic regression analysis showed that urinary catheter indwelling remained the most important risk factor (odds ratio (OR) 3.2; Table 3). Other risk factors were also related to UTI with different ORs, and the levels of Hb and Alb were inversely related to UTI risk.

## 4. Discussion

Female predominated in our study, which was compatible with the current trend of aging and gender distribution of the elderly in Taiwan. The overall UTI incidence in this study was higher than that in a previous study, possibly due to the enrollment of both outpatient and inpatient cases [11]. In this study, UTI diagnosis at the outpatient department largely depended on the judgment of clinicians by presence of clinical symptoms or signs, such as fever, pyuria, and/or bacteriuria, which might differ from person to person. In fact, the appropriate diagnostic criteria of UTI in the disabled and elderly remains unclear because most of the infections are asymptomatic and clinical signs are subtle and nonspecific [16,17]. Additionally, the mortality rate among our hospitalized patients is high (22.7%), indicating that UTI is a potentially life-threatening disease. On the other hand, some studies pointed that asymptomatic bacteriuria (ASB) is common among the elderly, with a prevalence ranging from 5% to 20% and even higher in long-term care residents [18]. Inappropriate antibiotic prescription may also have adverse effects, such as antibiotic-related diarrhea or Clostridium difficile infection [19]. Hence, differentiating ASB from true UTI is essential. Given the dilemma in treating UTI in the elderly, some articles have proposed an algorithm for UTI management (including the revised McGeer criteria targeting long-term care residents), but a universal consensus has not yet been reached [20,21].

Over 80% of the patients were fully urinary incontinent, and 50.3% of the total patients had chronic urinary catheter indwelling. Although our medical staff strictly complied with the principles of catheter care and nursing instructions were given to care givers, the urinary catheter remained the leading risk factor for UTI in this study (OR = 3.2), comparable to previous researches [10,11]. Prolonged urinary catheter placement is associated with inflammation of the bladder epithelium, bacterial colonization and biofilm formation [22]. Moreover, there was also a report disclosing the relationship between drug-resistant bacterial infection and catheterization [23]. Hence, avoiding unnecessary urinary catheter insertion and shortening the length of catheter placement as possible are crucial in preventing catheter-related UTI. Some experts have proposed protocols to guide appropriate catheter placement, and the UTI rate was successfully reduced under such evidence-based practice [24]. Besides, establishing a good nurse-caregiver partnership would also help to reduce the incidence of UTI [25]. Therefore, implementation of these protocols into our clinical practice should be considered in the future.

The Barthel index scores were inversely related to UTI risks. The result is compatible with previous studies that reported that the status of mobility and functional independence are protective factors against UTI and UTI-related mortality [26]. Among the multiple co-morbidities in this study, CAD (OR 1.79), DM (1.46), and CKD (1.54) were significantly related to UTI risks. Diabetic patients are susceptible to UTI due to multiple mechanisms, such as impaired immunity and autonomic neuropathy. Previous studies demonstrated the relationships both in general and home care populations [10]. Studies involving patients with CKD showed similar results [11,27]. We also found a relationship between hyper-polypharmacy and UTI. About one in five patients received over 10 prescriptions at the same time, which reflected the presence of multiple co-morbidities; this may predispose patients to infections. On the other hand, polypharmacy may also result in functional decline, frailty, incontinence, or even immunosuppression and mortality in the elderly [28]. Hence, avoiding unnecessary or potentially inappropriate medication (PIM) in the elderly remains an important concern. Using PIM-Taiwan explicit criteria could help clinicians to judge the necessity of prescriptions [29].

Under- or malnutrition has been proven to be associated with increased risk of infection, hospitalization, and mortality in the elderly; hence, assessment of nutritional status is an important concern in a home care population [30]. The MNA scale is an easy-to-perform and well-validated screening tool for nutrition in the elderly including home care population, and is recommended as part of comprehensive geriatric assessment [31]. MNA has also been reported to be related to UTI risk [32]. However, no significant difference was found in this study. This result was possibly due to the poor nutritional status of most patients, with only four patients who were adequately nourished, which could not reflect the difference between groups. 

Using albumin as predictor for nutrition remains debatable despite its ease of measurement in clinical practice [33]. As a marker of protein malnutrition, numerous studies have demonstrated its relationship with length of hospital stay, readmission rate, and morbidity and mortality [34]. Similarly, anemia also plays a role in infection [35]. A recent meta-analysis suggested that several blood biomarkers, including albumin and hemoglobin, are useful for adult malnutrition evaluation [36]. In the present study, hemoglobin and albumin levels showed significant difference between groups. However, since hypoalbuminemia is an acute phase reactant, the causal relationships require further evaluation. In view of the popularity of anemia in the elderly, the appropriate cut-off points remain unknown. Based on our findings, hemoglobin and albumin levels could be used for UTI risk assessment, and improving nutritional status should be taken as an infection prevention strategy.

The main results of this study resemble those studies focusing residents in facilities. Compared with a similar study conducted in Taiwan, our population was older and had more co-morbidities, which is comparable to the trend of aging and change in care modalities [11]. Home care population is expected to rapidly increase in the future. Therefore, how to improve healthy life expectancy and quality of home care is a considerable issue of the current society.

### Strength and Limitation

Patient data were obtained from the database of a single tertiary center in Northern Taiwan, which could not represent home care patients nationwide. However, the included home care population was generally older and had more comorbidities compared with those in previous studies, which is compatible with the current trend of aging. Moreover, since this is a retrospective cohort study, the causal relationships between risk factors and outcome would be difficult to attribute. However, we found that most of our results were similar to previous studies. No urinary culture reports were available in the present study, and the diagnostic criteria for UTI varied between physicians at outpatient department. Therefore, some asymptomatic cases might be enrolled in our analysis, which could pose potential bias to our study. Nevertheless, given that the presentations of infections are always subtle and atypical, and the mortality rate is high among the elderly, timely clinical judgement is important to treat UTI as early as possible.

## 5. Conclusions

Based on the results of our study, with the aim to avert UTI, we could avoid unnecessary prescriptions and prolonged unindicated urinary catheter placement, as well as improve the nutrition condition of home care patients. The prevalence of ASB among the older adults is still high. To further ameliorate the accuracy of diagnosis, a consensus of UTI management in the elderly is needed, which would require future investigation.

## Figures and Tables

**Table 1 ijerph-16-00566-t001:** Basic characteristics of the study population (N = 598).

Variables	n (%) or Mean ± SD
**Age**	81.9 ± 11.3
**Sex-male**	236 (39.5)
** -female**	362 (60.5)
**Main care giver-family members**	318 (53.2)
** -Taiwanese carer**	15 (2.5)
** -foreign carer**	265 (44.3)
**BMI (kg/m^2^)**	22.0 ± 3.6
**Coronary artery disease**	109 (18.2)
**Chronic kidney disease**	267 (44.6)
**Dementia**	185 (30.9)
**Cerebrovascular disease**	301 (50.3)
**Parkinsonism**	98 (16.4)
**Hypertension**	478 (79.9)
**Diabetes mellitus**	278 (46.5)
**Hyperlipidemia**	209 (34.9)
**Congestive heart failure**	208 (34.8)
**Chronic obstructive pulmonary disease/asthma/respiratory failure**	123 (20.6)
**History of cancer**	97 (16.2)
**Hyper-polypharmacy (taking ≥ 10 prescriptions)**	125 (20.9)
**Conscious status-GCS 13–15**	292 (48.8)
** -GCS 9–12**	207 (34.6)
** -GCS ≤ 8**	99 (16.6)
**Nasogastric tube use**	379 (63.4)
**Indwelling urinary catheter**	301 (50.3)
**Barthel index scores-0–20 (totally dependent)**	489 (81.8)
** -21–60 (severely dependent)**	109 (18.2)
**Barthel index sub-score: bladder control-0**	509 (85.1)
** -5**	82 (13.7)
** -10**	7 (1.2)
**Barthel index sub-score: bowel control-0**	450 (75.3)
** -5**	128 (21.4)
** -10**	20 (3.3)
**Norton scale scores**	11.2 ± 2.7
**Nutritional status-MNA ≥ 24**	4 (0.8)
** -24 > MNA ≥ 17**	269 (56.9)
** -MNA< 17**	200 (42.3)
**Hemoglobin (mg/dl)**	10.9 ± 1.8
**Albumin (mg/dl)**	3.8 ± 0.6

SD = standard deviation; BMI = body mass index; GCS = Glasgow coma scale; MNA = mini-nutritional assessment.

**Table 2 ijerph-16-00566-t002:** Comparison of baseline characteristics between UTI ^#^ and non-UTI groups.

Variable *	UTI (−) n = 317 (53%)	UTI (+) n = 281 (47%)	*p* Value
**Age (years)**	81.6 ± 11.4	82.4 ± 11.2	0.603
**Sex male**	132 (41.6%)	104 (37%)	0.276
** female**	185 (58.4%)	177 (63%)	
**Main care giver-family member**	165 (52.1%)	153 (54.4%)	0.074
** -Taiwanese carer**	4 (1.3%)	11 (3.9%)	
** -foreign carer**	148 (46.7%)	117 (41.6%)	
**BMI (kg/m^2^)**	21.9 ± 3.4	22.2 ± 3.9	0.27
**Coronary artery disease**	45 (14.2%)	64 (22.8%)	0.008
**Chronic kidney disease**	125 (39.4%)	142 (50.5%)	0.007
**Dementia**	94 (29.7%)	91 (32.4%)	0.48
**Cerebrovascular disease**	160 (50.5%)	141 (50.2%)	1
**Parkinsonism**	50 (15.8%)	48 (17.1%)	0.74
**Hypertension**	250 (78.9%)	228 (81.1%)	0.54
**Diabetes mellitus**	133 (42%)	145 (51.6%)	0.021
**Hyperlipidemia**	107 (33.8%)	102 (36.3%)	0.548
**Congestive heart failure**	100 (31.5%)	108 (38.4%)	0.086
**Chronic obstructive pulmonary disease/asthma/respiratory failure**	62 (19.6%)	61 (21.7%)	0.544
**History of cancer**	51 (16.1%)	46 (16.4%)	1
**Hyper-polypharmacy (taking ≥ 10 prescriptions)**	51(16.1%)	74 (26.3%)	0.002
** Conscious status-GCS 13–15**	154 (48.6%)	138 (49.1%)	0.99
** -GCS 9–12**	110 (34.7%)	97 (34.5%)	
** -GCS ≤ 8**	53 (16.7%)	46 (16.4%)	
**Nutritional status-MNA ≥ 24**	2 (0.8%)	2 (0.9%)	0.306
** -24 > MNA ≥ 17**	130 (53.5%)	139 (60.4%)	
** -MNA < 17**	111 (45.7%)	89 (38.7%)	
**Nasogastric tube use (%)**	197 (62.1%)	182 (64.8%)	0.506
**Urinary catheter indwelling (%)**	117 (36.9%)	184 (65.5%)	<0.001
**Barthel index scores-0–20**	248 (78.2%)	241 (85.8%)	0.017
** -21–60**	69 (21.8%)	40 (14.2%)	
**Barthel index bladder control-0**	250 (78.9%)	259 (92.2%)	<0.001
** -5**	62 (19.6%)	20 (7.1%)	
** -10**	5 (1.6%)	2 (0.7%)	
**Barthel index bowel control-0**	235 (74.1%)	215 (76.5%)	0.52
** -5**	69 (21.8%)	59 (21.0%)	
** -10**	13 (4.1%)	7 (2.5%)	
**Norton scale**	11.3 ± 2.9	11.0 ± 2.4	0.2
**Hemoglobin (mg/dl)**	11.1 ± 1.9	10.8 ± 1.7	0.024
**Albumin (mg/dl)**	3.9 ± 0.6	3.8 ± 0.6	0.002

^#^ The study population was divided into 2 groups: those who had at least one UTI episode during the observation period were designated as UTI (+) group, and those who never had such infection were UTI (−) group. A UTI episode during the observation period was defined by either admission for UTI treatment or visit to outpatient department for UTI and receiving antibiotic prescription. UTI diagnosis was confirmed by ICD-10-CM codes recorded in medical documents. For more information, please refer to the Material and Methods section. * Data are presented as number (percent) or mean ± SD.; SD = standard deviation; BMI = body mass index; GCS = Glasgow coma scale; MNA = mini-nutritional assessment.

**Table 3 ijerph-16-00566-t003:** Age and sex-adjusted multiple logistic regression analysis of risk factors for UTI.

Variable	OR	95% CI	*p* Value
**Urinary catheter indwelling**	3.21	2.30–4.50	<0.001
**Coronary artery disease**	1.79	1.17–2.73	0.007
**Chronic kidney disease**	1.54	1.11–2.14	0.011
**Diabetes mellitus**	1.46	1.05–2.02	0.024
**Hyper-polypharmacy**	1.84	1.23–2.75	0.003
**Barthel index scores**	1.62	1.04–2.53	0.032
**Albumin**	0.68	0.52–0.89	0.004
**Hemoglobin**	0.91	0.82–0.995	0.040

OR = odds ratio; CI = confidence interval.

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
