# Peer review of "Incidence and Risk Factors for Urinary Tract Infection in an Elder Home Care Population in Taiwan: A Retrospective Cohort Study"

_ijerph, 2019, doi:10.3390/ijerph16040566_

Round 1

Reviewer 1 Report

This is the report of clinical risk factors for UTI for elderly patients in home care population from Taiwan.

This is an important study in an epoch of aging society but there are several issues to be concerned.

Title needs to be revised and preferred to add “elder” or related words.

Introduction should be more focused on problem in the aging patients, and please consider to add the difference between elderly patients and younger ones for not only UTI but also general status.

What is the catheter role in this study? If you exclude the catheterized patients, what is the results?

What is UTI diagnosis standard in this study?

It needs more description as to the method of multivariate analysis.

Results

How about the immune-status or immune-related diseases or comorbidities?

Steroids or DM or chemotherapies?

The gender distribution is close to the standard or average?

What factors you saw as significant is cause for or results from UTI? And how about the comparison with the previous literatures?

How about PS?

Discussion

Do this patients’ category include asymptomatic ones?

How about the issue of urine catheter? What is the impact on this study?

Barthel index scores needs more explanation for better understanding.

Author Response

Response to Reviewer 1 Comments

Point 1: Title needs to be revised and preferred to add “elder” or related words.

Response 1: Thank you very much for the excellent suggestion. We have added the word “elder” in our title, so the name of our article has been changed into “Incidence and Risk Factors for Urinary Tract Infection in an Elder Home Care Population in Taiwan: A Retrospective Cohort Study”.

Point 2: Introduction should be more focused on problem in the aging patients, and please consider to add the difference between elderly patients and younger ones for not only UTI but also general status.

Response 2: Thank you very much for the meticulous review. The ageing process would lead to dramatic physiological changes, so the problems the elderly would face are quite different from the younger ones. Compared to young people, the elderly is more likely to suffer from various special physical and mental health challenges, such as falls, depression, delirium, frailty, pressure sores, and infections. We have listed some of the main problems which would be encountered by the elderly and also a brief comparison with the younger ones. The changes are showed in line 52-54, page 2.

Point 3: What is the catheter role in this study? If you exclude the catheterized patients, what is the results?

Response 3: Thank you very much for this question. Catheterized patients were included in our study and accounted for 50.3% of all subjects. Urinary catheter is a very common medical treatment in the home care population because the clients generally have poor bladder control. We included the catheterized home care patients with the aim to investigate whether the catheter would predispose this population to urinary tract infection. And the study result has confirmed the role of the catheter as one of the most important risk factor for UTI in this population.

Point 4: What is UTI diagnosis standard in this study?

Response 4: Thank you very much for this question. A UTI episode in our study is defined by any one of the following: (1) admission for UTI treatment, (2) a visit to the outpatient department for UTI and receiving antibiotics prescription. The diagnosis is confirmed by ICD-10 codes (such as urinary tract infection (N39.0), acute cystitis/acute cystitis without hematuria (N30.0), acute cystitis with hematuria (N30.01), other cystitis without hematuria (N30.8), other cystitis with hematuria (N30.81), cystitis, unspecified without hematuria (N30.9), cystitis, unspecified with hematuria (N30.91), and acute pyelonephritis (N10)) documented in the medical records. Hence, antibiotics treatment is required in either one of the conditions. The details of the diagnosis standard is described in line 95, page 2 in the subsection 2.3 measures of material and method.

Point 5: It needs more description as to the method of multivariate analysis.

Response 4: Thank you for your reminder. All the variables in the multiple logistic regression analysis were adjusted by age and sex. We have added the description in the context in line 194, page 3, line 218, page 3 and also the title of table 3.

Point 6: How about the immune-status or immune-related diseases or comorbidities? Steroids or DM or chemotherapies

Response 6: Thank you for your attention to this issue. The immune-related comorbidities obtained from the database were DM and cancer history. The cancer history might indicate that the patient had cancer in the past and the treatment courses had been completed.

As for DM, previous studies have demonstrated that DM patients are prone to UTI. The proposed explanations of this relationship are high prevalence rate of DM neuropathy and neurogenic bladder as well as a decrease in immune status. The findings of our study are compatible with those in previous literatures. (The relationship between DM and immune status was described in the context in line 296-298, page 7.)

Although there was a high prevalence rate of poly-prescriptions, the medication the clients receiving were mainly for common underlying diseases such as HTN, DM, CAD. Hence most of the long-term prescriptions did not include steroids.

Point 7: The gender distribution is close to the standard or average?

Response 7: Thank you for your query about this issue. Female predominates in our study population as 63%. According to the statistics of Taiwan government, women continue to outnumber men in the older ages, including those age 80 years and older. Hence, we consider the gender distribution in our study is close to the average. 

Point 8: What factors you saw as significant is cause for or results from UTI? And how about the comparison with the previous literatures?

Response 8: Thank you for your question. We considered urinary catheter, CAD, CKD, DM, polypharmacy and low Barthel index scores as causes for UTI. Though there were some difference in the variables, the main results of our study were similar to previous 2 studies conducted in home care patients in Taiwan and Finland respectively. The context is in line 262, page 6 and line 299, page 7.  

Point 9: How about PS?

Response 9: We are really sorry but we could not figure out what PS stands for. We are not sure if it is the initial of “performance status”. If so, according to the regulation of NHI program in Taiwan, only patients with limited ability of self-care (which means ECOG performance score 3 or more or Barthel index score 60 or less) are eligible for home care services. We further divided the Barthel index scores into 2 groups: score 20-60 and score less than 20. And the study result showed that the lower the Barthel index score, the higher the risk for UTI. We hope this is the correct answer to your question.

Point 10: Do this patients’ category include asymptomatic ones?

Response 10:  Yes. Given that the presentations of infections are always subtle and atypical, and the mortality rate is high, the UTI diagnosis at outpatient department mainly based on the judgment of experienced physicians. Therefore, some asymptomatic cases might be enrolled in our analysis, which could pose potential bias to our study. After thoroughly reviewing of literatures, there is no universal consensus about the UTI diagnosis in elderly. Some experts have proposed algorithms for diagnosing and managing UTI in this population, and we will consider adopting these protocols into our future studies.

Point 11: How about the issue of urine catheter? What is the impact on this study?

Response 11:  Urinary catheter was the main risk factor for UTI in our study, and about half of our patients were catheterized because of poor bladder control. Given the fact that the catheter could result in inflammation of bladder epithelium, bacterial colonization, and biofilm formation, which would be the pathogenesis for UTI, we should avoid unnecessary catheter placement and shorten the catheter-day as possible as we could.

We have added more descriptions about this issue, as shown in line 259-268, page 6.

Point 12: Barthel index scores needs more explanation for better understanding.

Response 12:  The Barthel index score represents the ability of a patient to perform activities of daily living and is commonly used for assessing functional status. It contains 10 domains as following: feeding, bathing, grooming, dressing, bowel control, bladder control, toilet use, transfers, mobility and climbing stairs. In Taiwan, a Barthel index score less than 60 is one of the enrolment criteria for home care services. We have added some narration regarding the domains contained in the Barthel index scores in line 180-182, page 3.

Reviewer 2 Report

Comments:

1.       Overall the introduction section is good. However, the articulation can be further enhanced by add information/data about the home health care sector in Taiwan, such as N of patient receiving home health care, projected growth.

2.       Please be specific and explicitly state that the research cited on page 2 line 45-48 is a study of the home health care system in the United States. Also, short observation period is not a strong argument why this study is necessary. Home health care is the most varied health sector cross nations. For example, a standard CMS payment period for a home health care episode is 60 days in the US.

3.       As aforementioned the significant variation in home health care systems across nations, it would be helpful to briefly introduce the home health care system in Taiwan. Such introduction will be also helpful to understanding the rationales behind the inclusion/exclusion criteria. For example the inclusion criterion of “receiving home care services for at least 3 months,” which may exclude many patient receiving home health care if in the context of e.g., the US home health care system. Otherwise, this criterion raises concerns of biased sample (i.e., a sample of patients with severe clinical conditions)

4.       Given that urinary catheter indwelling is a significant risk, the current indicator of chronic urinary catheter indwelling could miss a decent amount of patients that had urinary catheter indwelled but <30 days. Suggest to include an indicator of whether a patient had urinary catheter indwelling during the study period (including those with the catheter indwelling for <30 days) or not.

5.       Would suggest restructure the Materials and Methods section by organizing the content into subsections with appropriate subheadings, e.g., home health care in Taiwan, Data source, Measures, Analysis

6.       In the regression models, the variable of Barthel index urinary control was not included in the multivariable model without explanation, although it is significant in the binary analyses.

7.       In the Discussion section, the first paragraph is more appropriate for the Introduction section, and thus suggest remove it. Overall, the current discussion is limited and superficial. A more in depth discussion regarding the main findings is needed. For example, given the findings that urinary catheter indwelling and hyper-polypharmacy are two main risk factors of UTI, the authors may discuss what the current practice standards/protocols/guidelines are on urinary catheter utilization and medication prescription; and proposal some specific suggestions to improve the current practice in order to reduce UTI among home health care patients.

Author Response

Response to Reviewer 2 Comments

Point 1: Overall the introduction section is good. However, the articulation can be further enhanced by add information/data about the home health care sector in Taiwan, such as N of patient receiving home health care, projected growth

Response 1: Thank you very much for raising concern about this issue. We have searched the statistics released by the government, and found that there were about 100,000 patients receiving home care services in the year of 2017. In fact, there were about 900,000 people age 65 years or older require assistance in at least one domain of ADL or IADL. Besides, the Taiwan government has launched the Long-Term Care Act 2.0 in 2017, which greatly expands the array of services provided by the government to care for the elderly, hence we expect the home care population will grow rapidly in the near future.

The information regarding current home care sector is presented in the context from line 36, page 1 to line 50, page 2.

Point 2: Please be specific and explicitly state that the research cited on page 2 line 45-48 is a study of the home health care system in the United States. Also, short observation period is not a strong argument why this study is necessary. Home health care is the most varied health sector cross nations. For example, a standard CMS payment period for a home health care episode is 60 days in the US.

Response 2: Thank you very much for this suggestion and reminder. We have specified the nation where the study was conducted. And, since there’s great variation between home care systems in different countries, we added descriptions regarding Taiwan’s home care system both in the Introduction and Material and Method sections.

In Taiwan, only those with limited ability of self-care are eligible for home care services, hence the disabled elderly are the main beneficiaries of current home care system. Besides, with the universal reimbursement of national health insurance (NHI) program of Taiwan, the beneficiaries are able to receive the home care services as long as the nursing care needs are present. Therefore, the observation period of our study is relatively longer than the studies conducted in other countries. 

And the narration has been changed as shown in line 61-62, line 66.

Point 3: As aforementioned the significant variation in home health care systems across nations, it would be helpful to briefly introduce the home health care system in Taiwan. Such introduction will be also helpful to understanding the rationales behind the inclusion/exclusion criteria. For example the inclusion criterion of “receiving home care services for at least 3 months,” which may exclude many patient receiving home health care if in the context of e.g., the US home health care system. Otherwise, this criterion raises concerns of biased sample (i.e., a sample of patients with severe clinical conditions)

Response 3: Thank you for your excellent question. As stated in point 2, with the reimbursement of NHI program, the follow-up time could be relatively longer. With the aim to discover the risk factors that would be encountered during the home care services period rather than acute disease stages, we chose receiving at least 3 months of services as the inclusion criteria.

As the answer to point 2, we have added a paragraph describing the home care system in Taiwan in line 72-76, page 2. And thank you again for your advice.

Point 4: Given that urinary catheter indwelling is a significant risk, the current indicator of chronic urinary catheter indwelling could miss a decent amount of patients that had urinary catheter indwelled but <30 days. Suggest to include an indicator of whether a patient had urinary catheter indwelling during the study period (including those with the catheter indwelling for <30 days) or not.

Response 4: Thank you for the question. Most of our patients were totally incontinent with long-term urinary catheter indwelling, and most of them failed to remove the catheter due to underlying diseases (such as neurogenic bladder or cerebrovascular diseases). If the catheter was successfully removed, mostly it was done in the first 3 months of entry of home care services (which means the catheter was placed during acute stage of diseases following discharge). We would like to know the impact of long-term catheter placement, so this cut-off point was used.

Point 5: Would suggest restructure the Materials and Methods section by organizing the content into subsections with appropriate subheadings, e.g., home health care in Taiwan, Data source, Measures, Analysis.

Response 5: Thank you for the suggestion. We have restructured the Material and Methods section into 4 subsections: Home care system in Taiwan, Participants, Measures, and Data analysis.

Point 6: In the regression models, the variable of Barthel index urinary control was not included in the multivariable model without explanation, although it is significant in the binary analyses.

Response 6: Since about 99% of our patients in both groups had poor bladder control, we thought this difference did not worth further discussion in the following context. Besides, we used the Barthel index scores to represent the general functional status of the patients, hence we consider the bladder control variable could be omitted.

Point 7: In the Discussion section, the first paragraph is more appropriate for the Introduction section, and thus suggest remove it. Overall, the current discussion is limited and superficial. A more in depth discussion regarding the main findings is needed. For example, given the findings that urinary catheter indwelling and hyper-polypharmacy are two main risk factors of UTI, the authors may discuss what the current practice standards/protocols/guidelines are on urinary catheter utilization and medication prescription; and proposal some specific suggestions to improve the current practice in order to reduce UTI among home health care patients.

Response 7: Thank you for your suggestion. The first paragraph of the Discussion section was removed. Of the results of our study, after excluding the unmodifiable variables such as underlying diseases and baseline Barthel index scores, polypharmacy and urinary catheter placement are the main risk factors for UTI. Many experts have noticed the adverse effects of longterm catheter placement and have brought up some standardized protocols to improve the quality of catheter care as well as to avert unnecessary catheter placement. We consider adopting the protocols into our clinical practices. About the polypharmacy, we think clinicians should be more alert with the potentially inappropriate prescriptions and avoid such medications. We have added related descriptions in the context in line 266-268, page 6, and line 304-306, page 7.

Round 2

Reviewer 1 Report

Thank you for the comments and the revision.

However, the authors showed the responses to the reviewers’ comments only appeared in this letter or references in several ones. Please consider to reflect on the text and tables as well.

You can add more in study limitation.

Please consider to add the related references of Catheter associated UTI in the same area.

A nurse-family partnership intervention to increase the self-efficacy of family caregivers and reduce catheter-associated urinary tract infection in catheterized patients.

Int J Nurs Pract. 2015 Dec;21(6):771-9. doi: 10.1111/ijn.12319. Epub 2014 Apr 22.

Emergence of extended-spectrum β-lactamase-producing Escherichia coli in catheter-associated urinary tract infection in neurogenic bladder patients.

Am J Infect Control. 2014 Mar;42(3):e29-31. doi: 10.1016/j.ajic.2013.11.018.

Author Response

Response to Reviewer 1 Comments (round 2)

Point 1: However, the authors showed the responses to the reviewers’ comments only appeared in this letter or references in several ones. Please consider to reflect on the text and tables as well.

Response 1: Thank you very much for the reminder. About the eligibility of home care system in Taiwan, only those who are bed- or chair-bound over 50% while awake are qualified for applying the services. We have modified the descriptions for better understanding. (The changes could be found in line 76-78, page 2, subsection 2.1 Home care system in Taiwan).

And about the gender distribution of the elderly in Taiwan, the ratio of the male/female in our study population resembled the current trend of ageing: women outnumber men. We have added related description in line 244-245, page 6.

The variables in the multiple logistic regression analysis in table 3 were all adjusted by age and sex. We have modified the title of table 3 into “Age and sex-adjusted multiple logistic regression analysis of risk factors for UTI”. (The changes could be found in line 241, page 6).

And thank you again for your friendly reminder. 

Point 2: You can add more in study limitation.

Response 2: Thank you very much for the suggestion. One of the main limitations of our study was that the diagnostic criteria of UTI in outpatient department varied between physicians. Hence some patients who had asymptomatic bacteriuria would be enrolled in our analysis, which would pose potential bias to our study. Nevertheless, given that the presentations of infections are always subtle and atypical, and the mortality rate is high among the elderly, timely clinical judgement is quite important to treat UTI as early as possible. We have revised the limitations section of our study. (The changes could be found in line 339-360, page 7 to page 8.)

Point 3: Please consider to add the related references of Catheter associated UTI in the same area.

A nurse-family partnership intervention to increase the self-efficacy of family caregivers and reduce catheter-associated urinary tract infection in catheterized patients.

Int J Nurs Pract. 2015 Dec;21(6):771-9. doi: 10.1111/ijn.12319. Epub 2014 Apr 22.

Emergence of extended-spectrum β-lactamase-producing Escherichia coli in catheter-associated urinary tract infection in neurogenic bladder patients.

Am J Infect Control. 2014 Mar;42(3):e29-31. doi: 10.1016/j.ajic.2013.11.018.

Response 3: Thank you for your great advice. Since most of our patients are catheterized, the issues regarding CAUTI are important. In spite of exploring the pathogenesis and risk factors for UTI, appropriate diagnosis and management of the CAUTI are also quite essential. To lessen the CAUTI rate in home care population, adequate nursing instructions and a good nurse-family partnership are crucial. Besides, the catheter is also associated with the drug-resistant bacterial strain infection, so researching this relationship would be an important issue in the future.

The related articles have been added into the reference list as No. 23 and 25, and the descriptions were added in line 265, page 6 to line 290, page 7 and line 294-295, page 7.